# The Lorenz Curve: A Proper Framework to Define Satisfactory Measures of Symbol Dominance, Symbol Diversity, and Information Entropy

**DOI:** 10.3390/e22050542

**Published:** 2020-05-13

**Authors:** Julio A. Camargo

**Affiliations:** Unidad Docente de Ecología, Departamento de Ciencias de la Vida, Universidad de Alcalá, 28805 Alcalá de Henares, Madrid, Spain; julio.camargo@uah.es

**Keywords:** symbol dominance, symbol diversity, information entropy, Lorenz curve, Camargo statistics, Shannon’s entropy, Rényi’s entropy

## Abstract

Novel measures of symbol dominance (*d_C_*_1_ and *d_C_*_2_), symbol diversity (*D_C_*_1_ = *N* (1 − *d_C_*_1_) and *D_C_*_2_ = *N* (1 − *d_C_*_2_)), and information entropy (*H_C_*_1_ = log_2_
*D_C_*_1_ and *H_C_*_2_ = log_2_
*D_C_*_2_) are derived from Lorenz-consistent statistics that I had previously proposed to quantify dominance and diversity in ecology. Here, *d_C_*_1_ refers to the average absolute difference between the relative abundances of dominant and subordinate symbols, with its value being equivalent to the maximum vertical distance from the Lorenz curve to the 45-degree line of equiprobability; *d_C_*_2_ refers to the average absolute difference between all pairs of relative symbol abundances, with its value being equivalent to twice the area between the Lorenz curve and the 45-degree line of equiprobability; *N* is the number of different symbols or maximum expected diversity. These Lorenz-consistent statistics are compared with statistics based on Shannon’s entropy and Rényi’s second-order entropy to show that the former have better mathematical behavior than the latter. The use of *d_C_*_1_, *D_C_*_1_, and *H_C_*_1_ is particularly recommended, as only changes in the allocation of relative abundance between dominant (*p_d_* > 1/*N*) and subordinate (*p_s_* < 1/*N*) symbols are of real relevance for probability distributions to achieve the reference distribution (*p_i_* = 1/*N*) or to deviate from it.

## 1. Introduction

Following the early use of Shannon’s [1] entropy (*H_S_*) by some theoretical ecologists during the 1950s [2,3,4], *H_S_* has been extensively used in community ecology to quantify species diversity. Ecologists have considered the relative abundance or probability of the *i*th symbol in a message or sequence of *N* different symbols whose meaning is irrelevant [1,5,6] as the relative abundance or probability of the ith species in a community or assemblage of *S* different species whose phylogeny is irrelevant (i.e., all species are considered taxonomically equally distinct) [4,7,8]. This use of *H_S_* implies that the concept of species diversity is directly related to the concept of information entropy, basically representing the amount of information or uncertainty in a probability distribution defined for a set of *N* possible symbols [1] or a set of *S* possible species [4]. *H_S_* takes values from 0 to log_2_
*N* or log_2_
*S* and is properly expressed in bits, but it can also be expressed in nats or dits (also called bans, decits, or *Hartleys*) if the natural logarithm or the decimal logarithm is calculated [1,4,5,6,7,8].

In recent decades, several ecologists have, however, claimed that *H_S_* is a unsatisfactory diversity index because species diversity actually takes values from 1 to *S* and is ideally expressed in units of species (i.e., in the same units as *S*). Keeping this perspective in mind, and only considering the number of different symbols as the number of different species and the relative abundances of symbols as the relative abundances of species, Hill [9] proposed the exponential form of Shannon’s [1] entropy (*H_S_*) and the exponential form of Rényi’s [10] second-order entropy (*H_R_*) = the reciprocal of Simpson’s [11] concentration statistic (*λ*) as better alternatives to quantify species diversity, thereby assuming that the amount of information or uncertainty in a probability distribution defined for a set of *S* possible species was mathematically equivalent to the logarithm of its related species diversity. Similarly, we can assume that the amount of information or uncertainty (expressed in bits) in a probability distribution defined for a set of *N* possible symbols is mathematically equivalent to the binary logarithm of its related symbol diversity. Additionally, we can assume that symbol dominance characterizes the extent of relative abundance inequality among different symbols, particularly between dominant and subordinate symbols, and that symbol diversity equals the number of different symbols (*N*) or maximum expected diversity in any given message with equiprobability.

On the basis of these working assumptions, I first use the Lorenz curve [12] as the key framework to assess symbol dominance, symbol diversity, and information entropy. The contrast between symbol dominance and symbol redundancy is also highlighted. Subsequently, novel measures of symbol dominance (*d_C_*_1_ and *d_C_*_2_), symbol diversity (*D_C_*_1_ and *D_C_*_2_), and information entropy (*H_C_*_1_ and *H_C_*_2_) are derived from Lorenz-consistent statistics that I had previously proposed to quantify dominance and diversity in community ecology [13,14,15,16,17] and landscape ecology [18]. Finally, Lorenz-consistent statistics (*d_C_*_1_, *d_C_*_2_, *D_C_*_1_, *D_C_*_2_, *H_C_*_1_, and *H_C_*_2_) are compared with *H_S_*-based and *H_R_*-based statistics (*d_S_*, *d_R_*, *D_S_*, *D_R_*, *H_S_*, and *H_R_*) to show that the former have better mathematical behavior than the latter when measuring symbol dominance, symbol diversity, and information entropy in hypothetical messages. In this regard, I recently found that the corresponding versions of *d_C_*_1_, *d_C_*_2_, *D_C_*_1,_ and *D_C_*_2_ exhibited better mathematical behavior than the corresponding versions of *d_S_*, *d_R_*, *D_S_*, and *D_R_* when measuring land cover dominance and diversity in hypothetical landscapes [18]. This better mathematical behavior was inherent to the compatibility of *d_C_*_1_ and *d_C_*_2_ with the Lorenz-curve-based graphical representation of land cover dominance [18].

The Lorenz curve [12] was introduced in the early twentieth century as a graphical method to assess the inequality in the distribution of income among the individuals of a population. Subsequently, this graphical method and Lorenz-consistent indices of income inequality, such as Gini’s [19,20] index and Schutz’s [21] index, have become popular in the field of econometrics (see reviews in [22,23]). More recently, owing to the increasing economic inequality during the present market globalization [24], some authors have supported the use of Bonferroni’s [25] curve and Zenga’s [26] curve and related indices to better assess poverty, as these inequality measures are oversensitive to lower levels of the income distribution [27,28,29]. To me, however, the Lorenz curve represents the best and most logical framework to define satisfactory indices of inequality (dominance) and associated measures of diversity or entropy.

## 2. Materials and Methods

### 2.1. Assessing Symbol Dominance, Symbol Diversity, and Information Entropy within the Framework of the Lorenz Curve

In econometrics, the Lorenz curve [12] is ideally depicted within a unit (1 × 1) square, in which the cumulative proportion of income (the vertical *y*-axis) is related to the cumulative proportion of individuals (the horizontal *x*-axis), ranked from the person with the lowest income to the person with the highest income. The 45-degree (diagonal) line represents equidistribution or perfect income equality. Income inequality may be quantified as the maximum vertical distance from the Lorenz curve to the 45-degree line of equidistribution if only differences in income between the rich and the poor are of interest (this measure being equivalent to the value of Schutz’s inequality index), or as twice the area between the Lorenz curve and the 45-degree line of equidistribution if differences in income among all of the individuals are of interest (this measure being equivalent to the value of Gini’s inequality index), with both measures exhibiting the same value whenever income inequality occurs only between the rich and the poor (see reviews in [22,23]; also see [18]). Therefore, in any given population with *M* individuals, income inequality takes a minimum possible value of 0 when every person has the same income (= total income/*M*, including *M* = 1) and a maximum possible value of 1 − 1/*M* when a single person has all the income and the remaining *M* − 1 people have none, as persons with no income can exist in a population.

If we assume that symbol dominance characterizes the extent of relative abundance inequality among different symbols, particularly between dominant and subordinate symbols, then the Lorenz-curve-based graphical representation of symbol dominance is given by the separation of the Lorenz curve from the 45-degree line of equiprobability, in which every symbol *i* has the same relative abundance (*p_i_* = 1/*N*, with *N* = the number of different symbols). This separation may be quantified as the maximum vertical distance from the Lorenz curve to the 45-degree line if only differences in relative abundance between dominant and subordinate symbols are of interest, or as twice the area between the Lorenz curve and the 45-degree line if differences in relative abundance among all symbols are of interest, with both measures giving the same value whenever relative abundance inequality occurs only between dominant and subordinate symbols.

In any given message with equiprobability the relative abundance of each different symbol equals 1/*N*, meaning a symbol may be objectively regarded as dominant if its probability (*p_d_*) > 1/*N* and as subordinate if its probability (*p_s_*) < 1/*N*. I had already used an equivalent method to discriminate between dominant and subordinate species [13,14,15,16,17] and between dominant and subordinate land cover types [18]. Thus, symbol dominance takes a minimum possible value of 0 when every different symbol has the same relative abundance (= 1/*N*, including *N* = 1), and approaches a maximum possible value of 1 – 1/*N* when a single symbol has a relative abundance very close to 1 and the remaining *N* −1 symbols have minimum relative abundances (>0), as symbols with no abundance or zero probability do not exist in a message.

In addition, if we assume that symbol diversity equals the number of different symbols or maximum expected diversity *(N*) in any given message with equiprobability (symbol dominance = 0 because *p_i_* = 1/*N*), then symbol diversity in any given message with symbol dominance > 0 must equal the maximum expected diversity minus the impact of symbol dominance on it; that is, symbol diversity = *N* – (*N* × symbol dominance) = *N* (1 – symbol dominance). This Lorenz-consistent measure of symbol diversity is a function of both the number of different symbols and the equal distribution of their relative abundances (i.e., symbol diversity is a probabilistic concept free of semantic attributes), taking values from 1 to *N* (maximum diversity if *p_i_* = 1/*N*) and being properly expressed in units of symbols. Therefore, symbol diversity/*N* = 1 – symbol dominance (i.e., symbol dominance triggers the inequality between symbol diversity and its maximum expected value).

It should also be evident that the reciprocal of symbol diversity refers to the concentration of relative abundance in the same symbol, and consequently may be regarded as a Lorenz-consistent measure of symbol redundancy = 1/(*N* (1 − symbol dominance)). This redundancy measure is a function of both the fewness of different symbols and the unequal distribution of their relative abundances, taking values from 1/*N* to 1 (maximum redundancy if *N* = 1). Thus, symbol dominance (relative abundance inequality among different symbols) and symbol redundancy are distinct concepts, although the value of the former affects the value of the latter.

Lastly, if we assume that information entropy is mathematically equivalent to the binary logarithm of its related symbol diversity, then the resulting Lorenz-consistent measure of information entropy = log_2_ (*N* (1 − symbol dominance)). This entropy measure takes values from 0 to log_2_
*N* (maximum entropy if *p_i_* = 1/*N*) and is properly expressed in bits, quantifying the amount of information or uncertainty in a probability distribution defined for a set of *N* possible symbols. Obviously, the degree of uncertainty attains a minimum value of 0 as symbol redundancy reaches a maximum value of 1.

### 2.2. Deriving Measures of Symbol Dominance, Symbol Diversity, and Information Entropy from Lorenz-Consistent Statistics

Following the theoretical approach of assessing symbol dominance, symbol diversity, and information entropy within the framework of the Lorenz curve, novel measures of symbol dominance (*d_C_*_1_ and *d_C_*_2_), symbol diversity (*D_C_*_1_ and *D_C_*_2_), and information entropy (*H_C_*_1_ and *H_C_*_2_) are derived from Lorenz-consistent statistics, which I had previously proposed to quantify species dominance and diversity [13,14,15,16,17] and land cover dominance and diversity [18]. In this derivation the number of different species or land cover types is considered as the number of different symbols, and the probabilities of species or land cover types are considered as the probabilities of symbols:(1)$$d_{C}{}_{1}=\sum_{{{}^{d}}^{=1}}^{{}_{L}}\left(p_{d}–\;1/N\right)=(\sum_{{{}^{d}}^{\neq s=1}}^{{{}_{G}}_{}}\left(p_{d}–p_{s}\right))/N,$$
(2)$$D_{C}{}_{1}=N-\left(N\times d_{C}{}_{1}\right)=N(1-d_{C}{}_{1})=N-\sum \left(p_{d}-p_{s}\right),$$
*H_C_*_1_ = log_2_*D_C_*_1_ = log_2_ (*N* − ∑(*p_d_* − *p_s_*)),(3)
(4)$$d_{C}{}_{2}=(\sum_{{{}^{i}}^{\neq j=1}}^{{}_{K}}(│p_{i}-p_{j}│)/N,$$
(5)$$D_{C}{}_{{}_{2}}=N-\left(N\times d_{C}{}_{2}\right)=N\left(1-d_{C}{}_{2}\right)=N-\sum │p_{i}-p_{j}│,$$
*H_C_*_2_ = log_2_*D_C_*_2_ = log_2_ (*N* − ∑│*p_i_* − *p_j_|*),(6)
where *N* is the number of different symbols or maximum expected diversity, *p_d_* > 1/*N* is the relative abundance of each dominant symbol, *p_s_* < 1/*N* is the relative abundance of each subordinate symbol, *p_i_* and *p_j_* are the relative abundances of two different symbols in the same message, *L* is the number of dominant symbols, *G* is the number of subtractions between the relative abundances of dominant and subordinate symbols, and *K* = *N* (*N* − 1)/2 is the number of subtractions between all pairs of relative symbol abundances.

The dominance statistic *d_C_*_1_ refers to the average absolute difference between the relative abundances of dominant and subordinate symbols (Equation (1)), with its value being equivalent to the maximum vertical distance from the Lorenz curve to the 45-degree line of equiprobability (see also [18]). Accordingly, the value of *D_C_*_1_ equals the number of different symbols minus the impact of symbol dominance (*d_C_*_1_) on the maximum expected diversity (Equation (2)). The binary logarithm of this subtraction is the associated measure of information entropy (Equation (3)).

Likewise, the dominance statistic *d_C_*_2_ refers to the average absolute difference between all pairs of relative symbol abundances (Equation (4)), with its value being equivalent to twice the area between the Lorenz curve and the 45-degree line of equiprobability (see also [18]). Accordingly, the value of *D_C_*_2_ equals the number of different symbols minus the impact of symbol dominance (*d_C_*_2_) on the maximum expected diversity (Equation (5)). The binary logarithm of this subtraction is the associated measure of information entropy (Equation (6)).

Despite the above dissimilarities between Lorenz-consistent statistics of symbol dominance, symbol diversity, and information entropy, *d_C_*_1_ = *d_C_*_2_ = 0, *D_C_*_1_ = *D_C_*_2_ = *N*, and *H_C_*_1_ = *H_C_*_2_ = log_2_
*N* if there is equiprobability (*p_i_* = 1/*N*, including *N* = 1); and *d_C_*_1_ = *d_C_*_2_ > 0, *D_C_*_1_ = *D_C_*_2_ < *N*, and *H_C_*_1_ = *H_C_*_2_ < log_2_
*N* whenever relative abundance inequality occurs only between dominant and subordinate symbols. In this regard, it is worth noting that *d_C_*_1_ is comparable to Schutz’s [21] index of income inequality (also known as the Pietra ratio or Robin Hood index) and *d_C_*_2_ is comparable to Gini’s [19,20] index of income inequality. In fact, Gini’s index and Schutz’s index take the same value whenever income inequality occurs only between the rich and the poor (see reviews in [22,23]; also see [18]). However, there is a particular difference between the measurement of symbol dominance (*d_C_*_1_ and *d_C_*_2_) and the measurement of income inequality (Schutz’s index and Gini’s index): income inequality can reach a maximum value of 1 − 1/*M* when a single person has all the income and the remaining *M* – 1 people have none (as individuals with no income are considered to measure income inequality), but symbol dominance can only approach a maximum value of 1 – 1/*N* when a single symbol has a relative abundance very close to 1 and the remaining *N* – 1 symbols have minimum relative abundances (as symbols with no abundance or zero probability cannot be considered to measure symbol dominance).

Additionally, because the reciprocal of symbol diversity refers to the concentration of relative abundance in the same symbol (as already explained in Section 2.1), two Lorenz-consistent statistics of symbol redundancy are *R_C_*_1_ = 1/*D_C_*_1_ and *R_C_*_2_ = 1/*D_C_*_2_. *R_C_*_1_ and *R_C_*_2_ take values from 1/*N* to 1 (maximum redundancy if *N* = 1), and therefore their mathematical behavior can considerably differ from the mathematical behavior of Gatlin’s [30] classical redundancy index (*R_G_* = 1 – *H_S_*/log_2_
*N*). Indeed, since *R_G_* takes a maximum value of 1 if *N* = 1 and a minimum value of 0 if *p_i_* = 1/*N* [30], *R_G_* should be regarded as a combination of redundancy and dominance (see also [15]).

### 2.3. Comparing Lorenz-Consistent Statistics with H_S_-Based and H_R_-Based Statistics

Lorenz-consistent statistics of symbol dominance (*d_C_*_1_ and *d_C_*_2_), symbol diversity (*D_C_*_1_ and *D_C_*_2_), and information entropy (*H_C_*_1_ and *H_C_*_2_) are compared with statistics based on Shannon’s [1] entropy (*H_S_*) and Rényi’s [10] second-order entropy (*H_R_*). More specifically, on the basis of Hill’s [9] proposals for measuring diversity and Camargo’s [17] proposals for measuring dominance, the *H_S_*-based and *H_R_*-based statistics are:(7)$$H_{S}=-\sum_{{{}^{i}}^{=1}}^{N}p_{i}\log_{2}p_{i},$$
(8)$$D_{S}{}_{}=2^{H}{}^{S},$$
(9)$$d_{S}{}_{}=1-D_{S}/N,$$
(10)$$H_{R}={\;\log}_{2}1/\sum_{{{}^{i}}^{=1}}^{{}_{N}}p_{i}{}^{2},$$
(11)$$D_{R}{}_{}=2^{H}{}^{R},$$
(12)$$d_{R}=1-D_{R}/N,$$
where *p_i_* is the relative abundance or probability of the *i*th symbol in a message or sequence of *N* different symbols.

Although *d_C_*_1_ = *d_C_*_2_ = *d_S_* = *d_R_* = 0, *D_C_*_1_ = *D_C_*_2_ = *D_S_* = *D_R_* = *N*, and *H_C_*_1_ = *H_C_*_2_ = *H_S_* = *H_R_* = log_2_
*N* whenever there is equiprobability, differences in mathematical behavior between Lorenz-consistent statistics and *H_S_*-based and *H_R_*-based statistics were examined by computing all these statistics for the ten probability distributions (I–X) described as hypothetical messages in Table 1. As we can see, the hypothetical message V is the primary or starting distribution, having two different symbols with probabilities of 0.6 and 0.4. From distribution V to I, the probabilities of all different symbols are successively halved by doubling their number, with the whole relative abundance of dominant symbols that must be transferred to subordinate symbols to achieve equiprobability remaining steady (= 0.1). From distribution V to X, only the probabilities of subordinate symbols are successively halved by doubling their number, with the whole relative abundance of dominant symbols that must be transferred to subordinate symbols to achieve equiprobability approaching the probability of the single dominant symbol (= 0.6). Accordingly, the degree of dominance in each dominant symbol is given by the positive deviation of its probability (*p_d_*) from the expected equiprobable value of 1/*N*, while the degree of subordination in each subordinate symbol is given by the positive deviation of its probability (*p_s_*) from 1/*N*. Thus, in each probability distribution or hypothetical message, symbol dominance = symbol subordination = the average absolute difference between the relative abundances of dominant and subordinate symbols (Equation (1)) = the whole relative abundance of dominant symbols that must be transferred to subordinate symbols to achieve equiprobability (*P_transfer_* values in Table 1).

In addition, disparities in mathematical behavior between Lorenz-consistent statistics and *H_S_*-based and *H_R_*-based statistics were examined by computing all these statistics for the ten probability distributions (XI–XX) described as hypothetical messages in Table 2, where differences in relative abundance or probability occur not only between dominant and subordinate symbols (as in Table 1), but also among dominant symbols and among subordinate symbols. However, because the *P_transfer_* value equals 0.25 in all hypothetical messages, only changes in the allocation of relative abundance between dominant and subordinate symbols (but not among dominant symbols or among subordinate symbols) seem to be of real significance for probability distributions to achieve the reference distribution (involving equiprobability) or to deviate from it. The reasons for this are evident: in the case of a dominant symbol increasing its relative abundance at the expense of other dominant symbols (Table 2, relative abundances *p*_1_–*p*_5_ in probability distributions XVI–XIX), the resulting proportional abundance of all the dominant symbols is the same as before the transfer, since the increase in the probability of a dominant symbol (becoming more dominant) is compensated by an equivalent decrease in the probability of other dominant symbols (becoming less dominant); similarly, in the case of a subordinate symbol increasing its relative abundance at the expense of other subordinate symbols (Table 2, relative abundances *p*_6_–*p*_10_ in probability distributions XII–XV), the resulting proportional abundance of all the subordinate symbols is the same as before the transfer, since the increase in the probability of a subordinate symbol (becoming less subordinate) is compensated by an equivalent decrease in the probability of other subordinate symbols (becoming more subordinate or rare).

Probability distributions in Table 1 and Table 2 were selected to better assess differences in mathematical behavior between Lorenz-consistent statistics (Camargo’s indices) and *H_S_*-based and *H_R_*-based statistics. Otherwise, when using probability distributions that were chosen at random, we could obtain results that do not allow us to appreciate significant differences between the respective mathematical behaviors.

## 3. Results and Discussion

The Lorenz-curve-based graphical representation of symbol dominance (relative abundance inequality among different symbols) is shown in Figure 1. Estimated values of symbol dominance are 0.1 (I–V, with five Lorenz curves perfectly superimposed), 0.267 (VI), 0.4 (VII), 0.489 (VIII), 0.541 (IX), and 0.57 (X), with all these dominance values being equivalent to the respective Ptransfer values in Table 1. Additionally, estimated values of symbol diversity are 28.8 (I), 14.4 (II), 7.2 (III), 3.6 (IV), 1.8 (V), 2.199 (VI), 3.0 (VII), 4.599 (VIII), 7.803 (IX), and 14.19 (X) symbols, and estimated values of information entropy are 4.848 (I), 3.848 (II), 2.848 (III), 1.848 (IV), 0.848 (V), 1.137 (VI), 1.585 (VII), 2.202 (VIII), 2.964 (IX), and 3.827 (X) bits.

Differences in mathematical behavior between Lorenz-consistent statistics (*d_C_*_1_, *d_C_*_2_, *D_C_*_1_, *D_C_*_2_, *H_C_*_1_, and *H_C_*_2_) and *H_S_*-based and *H_R_*-based statistics (*d_S_*, *d_R_*, *D_S_*, *D_R_*, *H_S_*, and *H_R_*) are shown in Table 3. Because *d_C_*_1_, *d_C_*_2_, *D_C_*_1_, *D_C_*_2_, *H_C_*_1_, and *H_C_*_2_ are Lorenz-consistent, their estimated values match estimated values of symbol dominance, symbol diversity, and information entropy concerning Figure 1. In fact, estimated values of *d_C_*_1_ and *d_C_*_2_ are equivalent to the respective *P_transfer_* values in Table 1. By contrast, estimated values of *d_S_*, *d_R_*, *D_S_*, *D_R_*, *H_S_*, and *H_R_* do not match estimated values of symbol dominance, symbol diversity, and information entropy concerning Figure 1, while *d_S_* and *d_R_* exhibit values even greater than the upper limit for symbol dominance (= 0.6). Consequently, *D_S_* and *D_R_* can underestimate symbol diversity when differences in relative abundance between dominant and subordinate symbols are large or can overestimate it when such differences are relatively small.

The observed shortcomings in the measurement of symbol dominance (using *d_S_* and *d_R_*) and symbol diversity (using *D_S_* and *D_R_*) seem to be a consequence of the mathematical behavior of the associated entropy measures (*H_S_* and *H_R_*). As we can see in Table 3, from distribution V to I, where the *P_transfer_* value remains relatively small = 0.1 (Table 1), inequalities between entropy measures result in *H_S_* values > *H_R_* values > *H_C_*_1_ and *H_C_*_2_ values. On the contrary, from distribution VII to X, where the *P_transfer_* approaches a higher value of 0.6 (Table 1), inequalities between entropy measures result in *H_C_*_1_ and *H_C_*_2_ values > *H_S_* values > *H_R_* values. In fact, whereas the normalized entropies of *H_C_*_1_ and *H_C_*_2_ increase from distribution VII to X, the normalized entropies of *H_S_* and *H_R_* decrease markedly.

This remarkable finding would indicate that *H_C_*_1_ and *H_C_*_2_ can quantify the amount of information or uncertainty in a probability distribution more efficiently than *H_S_* and *H_R_*, particularly when differences between higher and lower probabilities are maximized by increasing the number of small probabilities (as shown in Table 3 regarding data in Table 1). After all, within the context of classical information theory, the information content of a symbol is an increasing function of the reciprocal of its probability [1,5,6,10] (also see [31,32]).

Other relevant disparities in mathematical behavior regarding measures of symbol dominance, symbol diversity, and information entropy are shown in Table 4. The respective values of *d_C_*_1_, *D_C_*_1_, and *H_C_*_1_ remain identical from distribution XI to XX, since *d_C_*_1_ is sensitive only to differences in relative abundance between dominant and subordinate symbols. On the other hand, because *d_C_*_2_ is sensitive to differences in relative abundance among all different symbols, the respective values of *d_C_*_2_, *D_C_*_2_, and *H_C_*_2_ do not remain identical from distribution XI to XX, even though they are equal in XII and XVI, in XIII and XVII, in XIV and XVIII, and in XV and XIX, as in each of these distribution pairs changes in the allocation of relative abundance among dominant symbols and among subordinate symbols are equivalent. A similar pattern of values is observed concerning *d_R_*, *D_R_*, and *H_R_*, but not regarding *d_S_*, *D_S_*, and *H_S_*, whose respective values remain distinct from distribution XI to XX.

## 4. Concluding Remarks

This theoretical analysis has shown that the Lorenz curve is a proper framework for defining satisfactory measures of symbol dominance, symbol diversity, and information entropy (Figure 1 and Table 3 and Table 4). The value of symbol dominance equals the maximum vertical distance from the Lorenz curve to the 45-degree line of equiprobability when only differences in relative abundance between dominant and subordinate symbols are quantified which is equivalent to the average absolute difference between the relative abundances of dominant and subordinate symbols = *d_C_*_1_ (Equation (1)) or equals twice the area between the Lorenz curve and the 45-degree line of equiprobability when differences in relative abundance among all symbols are quantified, which is equivalent to the average absolute difference between all pairs of relative symbol abundances = *d_C_*_2_ (Equation (4)). Symbol diversity = *N* (1 – symbol dominance) (i.e., *D_C_*_1_ = *N* (1 – *d_C_*_1_) and *D_C_*_2_ = *N* (1 – *d_C_*_2_)) and information entropy = log_2_ symbol diversity (i.e., *H_C_*_1_ = log_2_
*D_C_*_1_ and *H_C_*_2_ = log_2_
*D_C_*_2_). Additionally, the reciprocal of symbol diversity may be regarded as a satisfactory measure of symbol redundancy (i.e., *R_C_*_1_ = 1/*D_C_*_1_ and *R_C_*_2_ = 1/*D_C_*_2_).

This study has also shown that Lorenz-consistent statistics (*d_C_*_1_, *d_C_*_2_, *D_C_*_1_, *D_C_*_2_, *H_C_*_1_, and *H_C_*_2_) have better mathematical behavior than *H_S_*-based and *H_R_*-based statistics (*d_S_*, *d_R_*, *D_S_*, *D_R_*, *H_S_*, and *H_R_*), exhibiting greater coherence and objectivity when measuring symbol dominance, symbol diversity, and information entropy (Table 3 and Table 4). However, considering that the 45-degree line of equiprobability (Figure 1) represents the reference distribution (*p_i_* = 1/*N*), and that only changes in the allocation of relative abundance between dominant and subordinate symbols (but not among dominant symbols or among subordinate symbols) seem to have true relevance for probability distributions to achieve the reference distribution or to deviate from it (Table 2), the use of *d_C_*_1_, *D_C_*_1_, and *H_C_*_1_ may be more practical and preferable than the use of *d_C_*_2_, *D_C_*_2_, and *H_C_*_2_ in measuring symbol dominance, symbol diversity, and information entropy. In this regard, it should be evident that if the number of different symbols (*N*) is fixed in any given message, increasing differences in relative abundance between dominant and subordinate symbols necessarily imply decreases in symbol diversity and information entropy, whereas decreasing differences in relative abundance between dominant and subordinate symbols necessarily imply increases in symbol diversity and information entropy, with these two variables taking a maximum if *p_i_* = 1/*N*. By contrast, increasing or decreasing differences in relative abundance among dominant symbols or among subordinate symbols will not affect symbol diversity and information entropy, since the decrease or increase in the information content of a dominant or subordinate symbol is compensated by an equivalent increase or decrease in the information content of other dominant or subordinate symbols.

## Figures and Tables

**Figure 1 entropy-22-00542-f001:**
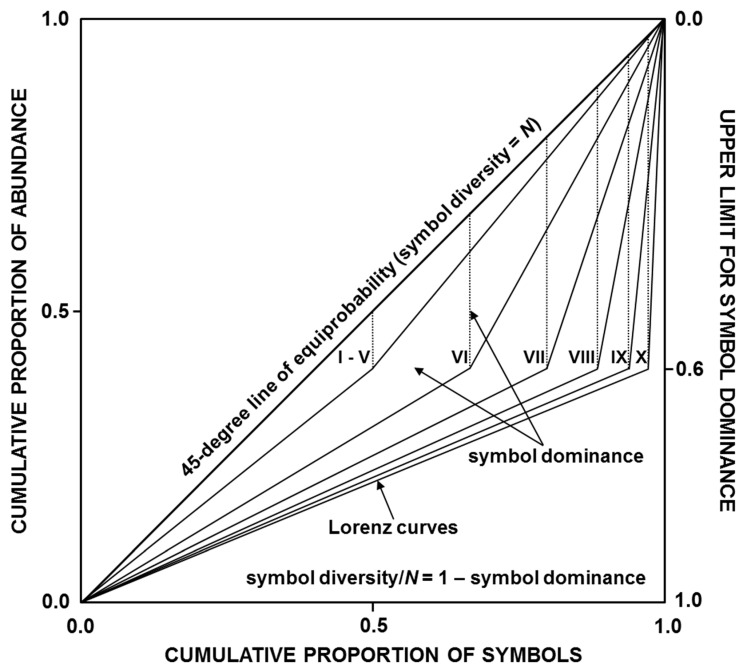
The cumulative proportion of abundance is related to the cumulative proportion of symbols, ranked from the symbol with the lowest relative abundance to the symbol with the highest relative abundance, for the ten probability distributions (I–X) described as hypothetical messages in Table 1. The reference distribution is depicted by the 45-degree line of equiprobability, where every symbol has the same relative abundance = 1/*N*, symbol dominance = 0, and symbol diversity = the number of different symbols (*N*). Symbol dominance may be estimated as the maximum vertical distance from the Lorenz curve to the 45-degree line, or as twice the area between the Lorenz curve and the 45-degree line, with both measures giving the same value whenever relative abundance inequality occurs only between dominant and subordinate symbols (as shown in this figure). In addition, symbol diversity = *N* (1 – symbol dominance), symbol redundancy = 1/symbol diversity, and information entropy = log_2_ symbol diversity.

**Table 1 entropy-22-00542-t001:** Ten probability distributions (I–X) are described as hypothetical messages: *N* = the number of different symbols; *p*_1_–*p*_33_ = the relative abundances of symbols (symbol probabilities); *P_transfer_* = the whole relative abundance of dominant symbols (*p_d_* > 1/*N*) that must be transferred to subordinate symbols (*p_s_* < 1/*N*) to achieve equiprobability (*p_i_* = 1/*N*, including *N* = 1).

	I	II	III	IV	V	VI	VII	VIII	IX	X
*N*	32	16	8	4	2	3	5	9	17	33
*p* _1_	0.0375	0.075	0.15	0.3	0.6	0.6	0.6	0.6	0.6	0.6
*p* _2_	0.0375	0.075	0.15	0.3	0.4	0.2	0.1	0.05	0.025	0.0125
*p* _3_	0.0375	0.075	0.15	0.2		0.2	0.1	0.05	0.025	0.0125
*p* _4_	0.0375	0.075	0.15	0.2			0.1	0.05	0.025	0.0125
*p* _5_	0.0375	0.075	0.1				0.1	0.05	0.025	0.0125
*p* _6_	0.0375	0.075	0.1					0.05	0.025	0.0125
*p* _7_	0.0375	0.075	0.1					0.05	0.025	0.0125
*p* _8_	0.0375	0.075	0.1					0.05	0.025	0.0125
*p* _9_	0.0375	0.05						0.05	0.025	0.0125
*p* _10_	0.0375	0.05							0.025	0.0125
*p* _11_	0.0375	0.05							0.025	0.0125
*p* _12_	0.0375	0.05							0.025	0.0125
*p* _13_	0.0375	0.05							0.025	0.0125
*p* _14_	0.0375	0.05							0.025	0.0125
*p* _15_	0.0375	0.05							0.025	0.0125
*p* _16_	0.0375	0.05							0.025	0.0125
*p* _17_	0.025								0.025	0.0125
*p* _18_	0.025									0.0125
*p* _19_	0.025									0.0125
*p* _20_	0.025									0.0125
*p* _21_	0.025									0.0125
*p* _22_	0.025									0.0125
*p* _23_	0.025									0.0125
*p* _24_	0.025									0.0125
*p* _25_	0.025									0.0125
*p* _26_	0.025									0.0125
*p* _27_	0.025									0.0125
*p* _28_	0.025									0.0125
*p* _29_	0.025									0.0125
*p* _30_	0.025									0.0125
*p* _31_	0.025									0.0125
*p* _32_	0.025									0.0125
*p* _33_										0.0125
*P_transfer_*	0.1	0.1	0.1	0.1	0.1	0.267	0.4	0.489	0.541	0.57

**Table 2 entropy-22-00542-t002:** Ten probability distributions (XI–XX) are described as hypothetical messages: *N* = the number of different symbols; *p*_1_–*p*_10_ = the relative abundances of symbols (symbol probabilities); *P_transfer_* = the whole relative abundance of dominant symbols (*p_d_* > 1/*N*) that must be transferred to subordinate symbols (*p_s_* < 1/*N*) to achieve equiprobability (*p_i_* = 1/*N*, including *N* = 1).

	XI	XII	XIII	XIV	XV	XVI	XVII	XVIII	XIX	XX
*N*	10	10	10	10	10	10	10	10	10	10
*p* _1_	0.15	0.15	0.15	0.15	0.15	0.19	0.19	0.19	0.19	0.19
*p* _2_	0.15	0.15	0.15	0.15	0.15	0.14	0.17	0.17	0.17	0.17
*p* _3_	0.15	0.15	0.15	0.15	0.15	0.14	0.13	0.15	0.15	0.15
*p* _4_	0.15	0.15	0.15	0.15	0.15	0.14	0.13	0.12	0.13	0.13
*p* _5_	0.15	0.15	0.15	0.15	0.15	0.14	0.13	0.12	0.11	0.11
*p* _6_	0.05	0.09	0.09	0.09	0.09	0.05	0.05	0.05	0.05	0.09
*p* _7_	0.05	0.04	0.07	0.07	0.07	0.05	0.05	0.05	0.05	0.07
*p* _8_	0.05	0.04	0.03	0.05	0.05	0.05	0.05	0.05	0.05	0.05
*p* _9_	0.05	0.04	0.03	0.02	0.03	0.05	0.05	0.05	0.05	0.03
*p* _10_	0.05	0.04	0.03	0.02	0.01	0.05	0.05	0.05	0.05	0.01
*P_transfer_*	0.25	0.25	0.25	0.25	0.25	0.25	0.25	0.25	0.25	0.25

**Table 3 entropy-22-00542-t003:** Measures of symbol dominance (*d_C_*_1_, *d_C_*_2_, *d_R_*, and *d_S_*), symbol diversity (*D_C_*_1_, *D_C_*_2_, *D_R_*, and *D_S_*), and information entropy (*H_C_*_1_, *H_C_*_2_, *H_R_*, and *H_S_*) are computed for the ten probability distributions (I–X) described as hypothetical messages in Table 1. *H_max_* = log_2_
*N* = maximum expected entropy; *H_C_*_1_/*H_max_*, *H_C_*_2_/*H_max_*, *H_R_*/*H_max_*, and *H_S_*/*H_max_* = normalized entropies. All statistics are explained in the text.

	I	II	III	IV	V	VI	VII	VIII	IX	X
*d_C_* _1_	0.100	0.100	0.100	0.100	0.100	0.267	0.400	0.489	0.541	0.570
*D_C_* _1_	28.800	14.400	7.200	3.600	1.800	2.199	3.000	4.599	7.803	14.190
*H_C_* _1_	4.848	3.848	2.848	1.848	0.848	1.137	1.585	2.202	2.964	3.827
*H_C_*_1_/*H_max_*	0.970	0.962	0.949	0.924	0.848	0.717	0.683	0.695	0.725	0.759
*d_C_* _2_	0.100	0.100	0.100	0.100	0.100	0.267	0.400	0.489	0.541	0.570
*D_C_* _2_	28.800	14.400	7.200	3.600	1.800	2.199	3.000	4.599	7.803	14.190
*H_C_* _2_	4.848	3.848	2.848	1.848	0.848	1.137	1.585	2.202	2.964	3.827
*H_C_*_2_/*H_max_*	0.970	0.962	0.949	0.924	0.848	0.717	0.683	0.695	0.725	0.759
*d_R_*	0.038	0.038	0.038	0.038	0.038	0.242	0.500	0.708	0.841	0.917
*D_R_*	30.768	15.384	7.692	3.846	1.923	2.273	2.500	2.632	2.703	2.740
*H_R_*	4.943	3.943	2.943	1.943	0.943	1.184	1.322	1.396	1.434	1.454
*H_R_*/*H_max_*	0.989	0.986	0.981	0.972	0.943	0.747	0.569	0.440	0.351	0.288
*d_S_*	0.020	0.020	0.020	0.020	0.020	0.138	0.317	0.500	0.650	0.762
*D_S_*	31.360	15.680	7.840	3.920	1.960	2.586	3.413	4.503	5.942	7.841
*H_S_*	4.971	3.971	2.971	1.971	0.971	1.371	1.771	2.171	2.571	2.971
*H_S_*/*H_max_*	0.994	0.993	0.990	0.985	0.971	0.865	0.763	0.685	0.629	0.589
*H_max_*	5.000	4.000	3.000	2.000	1.000	1.585	2.322	3.170	4.087	5.044

**Table 4 entropy-22-00542-t004:** Measures of symbol dominance (*d_C_*_1_, *d_C_*_2_, *d_R_*, and *d_S_*), symbol diversity (*D_C_*_1_, *D_C_*_2_, *D_R_*, and *D_S_*), and information entropy (*H_C_*_1_, *H_C_*_2_, *H_R_*, and *H_S_*) are computed for the ten probability distributions (XI–XX) described as hypothetical messages in Table 2. *H_max_* = log_2_
*N* = maximum expected entropy; *H_C_*_1_/*H_max_*, *H_C_*_2_/*H_max_*, *H_R_*/*H_max_*, and *H_S_*/*H_max_* = normalized entropies. All statistics are explained in the text.

	XI	XII	XIII	XIV	XV	XVI	XVII	XVIII	XIX	XX
*d_C_* _1_	0.250	0.250	0.250	0.250	0.250	0.250	0.250	0.250	0.250	0.250
*D_C_* _1_	7.500	7.500	7.500	7.500	7.500	7.500	7.500	7.500	7.500	7.500
*H_C_* _1_	2.907	2.907	2.907	2.907	2.907	2.907	2.907	2.907	2.907	2.907
*H_C_*_1_/*H_max_*	0.875	0.875	0.875	0.875	0.875	0.875	0.875	0.875	0.875	0.875
*d_C_* _2_	0.250	0.270	0.282	0.288	0.290	0.270	0.282	0.288	0.290	0.330
*D_C_* _2_	7.500	7.300	7.180	7.120	7.100	7.300	7.180	7.120	7.100	6.700
*H_C_* _2_	2.907	2.868	2.844	2.832	2.828	2.868	2.844	2.832	2.828	2.744
*H_C_*_2_/*H_max_*	0.875	0.863	0.856	0.852	0.851	0.863	0.856	0.852	0.851	0.826
*d_R_*	0.200	0.213	0.220	0.224	0.225	0.213	0.220	0.224	0.225	0.248
*D_R_*	8.000	7.870	7.800	7.760	7.750	7.870	7.800	7.760	7.750	7.520
*H_R_*	3.000	2.976	2.963	2.956	2.954	2.976	2.963	2.956	2.954	2.911
*H_R_*/*H_max_*	0.903	0.896	0.892	0.890	0.889	0.896	0.892	0.890	0.889	0.876
*d_S_*	0.122	0.137	0.149	0.157	0.161	0.128	0.131	0.133	0.134	0.173
*D_S_*	8.779	8.628	8.512	8.431	8.387	8.724	8.691	8.675	8.662	8.269
*H_S_*	3.134	3.109	3.089	3.076	3.068	3.125	3.120	3.117	3.115	3.048
*H_S_*/*H_max_*	0.943	0.936	0.930	0.926	0.924	0.941	0.939	0.938	0.937	0.918
*H_max_*	3.322	3.322	3.322	3.322	3.322	3.322	3.322	3.322	3.322	3.322

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
