# Peer review of "The Lorenz Curve: A Proper Framework to Define Satisfactory Measures of Symbol Dominance, Symbol Diversity, and Information Entropy"

_entropy, 2020, doi:10.3390/e22050542_

Round 1

Reviewer 1 Report

The manuscript introduces a measure for diversity, dominance and information entropy based on the Lorentz-curve analysis. I am not conviced, that similar measures are not used already. E.g. the manuscript does not mention the Gini index, which is a very widely used measure. There are several further measues that attribute diversity and information richness of a distribution.

Considering the use of Lorentz curves as a standard technique in several fields of applied sciences e.g. biology, economics or artificial intelligence, the author shuld give a much broader overview of available measures that lowers the chance of reinventing an already known index.

The essence of the manuscript, where the author claims to show the advantages of the new indices, is based on a single numerical study. Here, some ad hoc distributions are characterised by traditional and by the new measures.

Are these findings robust against variations of the datasets? There could be other datasets, where the new measures perform worse than earlier measures.

What are the circumstances, that should be satisfied, when the new measures perform better than others?

What are the weaknesses of the new measures? There is no measure, which cannot fail.

I have reported plagiarism, which is self-plagiarism: Figure 1 is taken (copied as it is) from an earlier publication of the same author.

I have assessed the topic as an extremely interesting topic, so the readers of Entropy would be interested in new results and new proposals for measuring dominance, diversity and information content of a sequence. But republishing an earlier result, without clear notice of this fact and just mentioning the original paper as a reference, is disappointing for an enthusiastic reader.

Author Response

Responses to the comments of reviewer #1:

1 - Reviewer’s comment:

“The manuscript introduces a measure for diversity, dominance and information entropy based on the Lorentz-curve analysis. I am not conviced, that similar measures are not used already. E.g. the manuscript does not mention the Gini index, which is a very widely used measure. There are several further measues that attribute diversity and information richness of a distribution.”

Author’s response:

As far as I know no previously published paper has related the three concepts of symbol dominance, symbol diversity and information entropy, and much less within the framework of the Lorenz curve. In this sense my submitted manuscript represents an original contribution, even though the proposed Lorenz-consistent statistics of symbol dominance (dC1 and dC2), symbol diversity (DC1 = N (1 – dC1) and DC2 = N (1 – dC2)) and information entropy (HC1 = log2 DC1 = log2 (N – ∑(pd – ps)) and HC2 = log2 DC2 = log2 (N – ∑│pi – pj│)) are derived from Lorenz-consistent statistics that I had previously proposed to quantify dominance and diversity in ecology (Camargo 1992a, 1992b, 1993, 1995, 2008, 2019). However, they are novel, particularly the indices of information entropy (HC1 = log2 (N – ∑(pd – ps)) and HC2 = log2 (N – ∑│pi – pj│)) in which N is the number of different symbols or maximum expected diversity, pd > 1/N is the relative abundance of each dominant symbol, ps < 1/N is the relative abundance of each subordinate symbol, and pi and pj are the relative abundances of two different symbols in the same message.

Also, because they represent a new approach to quantify information entropy, relating their values to the values o symbol dominance and symbol diversity, I have considered mandatory to compare the mathematical behavior of these Lorenz-consistent statistics of symbol dominance, symbol diversity and information entropy with the mathematical behavior of statistics based on the classical Shannon’s entropy (HS = – ∑ pi log2 pi) and Rényi`s (1961) second order entropy (HR = log2 1/∑ pi2).

On the other hand, I have included in the second version of the submitted manuscript (lines 70-74, 83-99, 186-197) additional information regarding the Lorenz curve and related indices of income inequality (such as the Gini index and the Schutz index) that are popular in the field of econometrics. In this regard it is worth noting that dC1 is comparable to the Schutz’s [21] index of income inequality (also known as Pietra ratio or Robin Hood index) and dC2 is comparable to the Gini’s [19, 20] index of income inequality. However, there is a particular difference between the measurement of symbol dominance (dC1 and dC2) and the measurement of income inequality (the Schutz’s index and the Gini’s index): income inequality can reach a maximum value of 1 – 1/M when a single person has all the income and the remaining M – 1 people have none (as individuals with no income are considered to measure income inequality), but symbol dominance can only approach a maximum value of 1 – 1/N when a single symbol has a relative abundance very close to 1 and the remaining N – 1 symbols have minimum relative abundances (as symbols with no abundance or zero probability cannot be considered to measure symbol dominance).

2 - Reviewer’s comment:

“Considering the use of Lorentz curves as a standard technique in several fields of applied sciences e.g. biology, economics or artificial intelligence, the author should give a much broader overview of available measures that lowers the chance of reinventing an already known index.”

Author’s response:

As already commented above, no previously published paper has related the three concepts of symbol dominance, symbol diversity and information entropy, and much less within the framework of the Lorenz curve. Moreover, as indicated in the ‘Introduction’ section of both versions of the submitted manuscript, novel measures of symbol dominance (dC1 and dC2), symbol diversity (DC1 = N (1 – dC1) and DC2 = N (1 – dC2)) and information entropy (HC1 = log2 DC1 = log2 (N – ∑(pd – ps)) and HC2 = log2 DC2 = log2 (N – ∑│pi – pj│)) are derived from Lorenz-consistent statistics that I had previously proposed to quantify dominance and diversity in community ecology (Camargo 1992a, 1992b, 1993, 1995, 2008) and landscape ecology (Camargo 2019).

3 - Reviewer’s comment:

“The essence of the manuscript, where the author claims to show the advantages of the new indices, is based on a single numerical study. Here, some ad hoc distributions are characterized by traditional and by the new measures.”

Author’s response:

Probability distributions in Tables 1 and 2 of the submitted manuscript were selected to better assess differences in mathematical behavior between Lorenz-consistent statistics (Camargo’s indices) and Lorenz-inconsistent statistics (indices based on Shannon’s entropy and Rényi’s second order entropy). Otherwise, in the case of using probability distributions chosen at random, we can obtain results that do not allow us to appreciate significant differences between the respective mathematical behaviors. This explanation has been incorporated at the end of the subsection 2.3 (Comparing Lorenz-Consistent Statistics with HS-Based and HR-Based Statistics) of the second (revised) version of the submitted manuscript (lines 288-292 of the revised manuscript with track changes).

With the use of those selected probability distributions I have really demonstrated the existence of significant differences in mathematical behavior, Lorenz-consistent statistics exhibiting greater coherence and objectivity than Lorenz-inconsistent statistics. In fact, as shown in Tables 3 and 4 regarding probability distributions described in Tables 1 and 2, the value of Camargo’s dominance statistic (dC1) always is equivalent to the whole relative abundance of dominant symbols that must be transferred to subordinate symbols to achieve equiprobability, that is, maximum coherence and objectivity because only changes in the allocation of relative abundance between dominant and subordinate symbols (but not among dominant symbols or among subordinate symbols) seem to have true relevance for probability distributions to achieve the reference distribution (involving equiprobability) or to deviate from it.

Furthermore, as shown in Table 3 regarding data in Table 1, whereas the normalized entropies of HC1 and HC2 increase from distribution VII to X, the normalized entropies of HS and HR decrease markedly. And as commented in the section 3 (Results and Discussion), this remarkable finding would indicate that HC1 and HC2 can quantify more efficiently than HS and HR the amount of information or uncertainty in a probability distribution, particularly when differences between higher and lower probabilities are maximized by increasing the number of small probabilities. After all, within the context of classical information theory, the information content of a symbol is an increasing function of the reciprocal of its probability.

In this regard I wish the reviewer had at least once highlighted the importance of that remarkable finding, but to my regret it has not been so.

4 - Reviewer’s comment:

“Are these findings robust against variations of the datasets? There could be other datasets, where the new measures perform worse than earlier measures”.

Author’s response:

To me the main attributes of a satisfactory measurement are coherence and objectivity. The theoretical analysis performed in the submitted manuscript has clearly shown that Lorenz-consistent statistics (dC1, dC2, DC1, DC2, HC1 and HC2) have better mathematical behavior than Lorenz-inconsistent statistics (dS, dR, DS, DR, HS and HR), exhibiting greater coherence and objectivity when measuring symbol dominance, symbol diversity and information entropy (as shown in Tables 3 and 4 of the submitted manuscript). However, considering that the 45-degree line of equiprobability (Figure 1) represents the reference distribution, and that only changes in the allocation of relative abundance between dominant and subordinate symbols (but not among dominant symbols or among subordinate symbols) seem to have true relevance for probability distributions to achieve the reference distribution or to deviate from it (as shown in Tables 1 and 2 of the submitted manuscript), the use of dC1, DC1 and HC1 may be more practical and preferable than the use of dC2, DC2 and HC2 to measure symbol dominance, symbol diversity and information entropy. This is my overall conclusion indicated in the section 4 (Concluding Remarks).

5 - Reviewer’s comment:

“What are the circumstances, that should be satisfied, when the new measures perform better than others?”

Author’s response:

Since Lorenz-consistent statistics (dC1, dC2, DC1, DC2, HC1 and HC2) exhibit greater coherence and objectivity than Lorenz-inconsistent statistics (dS, dR, DS, DR, HS and HR) when measuring symbol dominance, symbol diversity and information entropy (as shown in Tables 3 and 4 of the submitted manuscript), I think they can perform better in any circumstance. In fact, as shown in Table 3, Lorenz-inconsistent statistics (DS, DR, HS and HR) can underestimate symbol diversity and information entropy when differences in relative abundance between dominant and subordinate symbols are large or can overestimate them when such differences are relatively small.

However, as already commented above, the use of dC1, DC1 and HC1 may be more practical and preferable than the use of dC2, DC2 and HC2 to measure symbol dominance, symbol diversity and information entropy, since the value of dC1 always is equivalent to the whole relative abundance of dominant symbols that must be transferred to subordinate symbols to achieve equiprobability (i.e., maximum coherence and objectivity). Actually, as shown in Tables 1 and 2 of the submitted manuscript, only changes in the allocation of relative abundance between dominant and subordinate symbols (but not among dominant symbols or among subordinate symbols) seem to have true relevance for probability distributions to achieve the reference distribution (involving equiprobability) or to deviate from it.

The reasons for this fact are evident (as commented in lines 271-282 of the revised manuscript with track changes): in the case of a dominant symbol increasing its relative abundance at the expense of other dominant symbol (Table 2), the resulting proportional abundance of the whole dominant symbols is the same as before the transfer, since the increase in the probability of a dominant symbol (becoming more dominant) is compensated by an equivalent decrease in the probability of other dominant symbol (becoming less dominant); similarly, in the case of a subordinate symbol increasing its relative abundance at the expense of other subordinate symbol (Table 2), the resulting proportional abundance of the whole subordinate symbols is the same as before the transfer, since the increase in the probability of a subordinate symbol (becoming less subordinate) is compensated by an equivalent decrease in the probability of other subordinate symbol (becoming more subordinate or rare).

6 - Reviewer’s comment:

“What are the weaknesses of the new measures? There is no measure, which cannot fail.”

Author’s response:

In the case of dC2, DC2 (= N (1 – dC2)) and HC2 (= log2 DC2 = log2 (N – ∑│pi – pj│)) the only weakness, I think, would be that the value of dC2 is sensitive not only to differences in relative abundance between dominant and subordinate symbols but also to differences in relative abundance among dominant symbols and among subordinate symbols. This is the reason because I overall conclude that the use of dC1, DC1 (= N (1 – dC1)) and HC1 (= log2 DC1 = log2 (N – ∑(pd – ps))) may be more practical and preferable than the use of dC2, DC2 and HC2 to measure symbol dominance, symbol diversity and information entropy, since the value of dC1 always is equivalent to the whole relative abundance of dominant symbols that must be transferred to subordinate symbols to achieve equiprobability (i.e., maximum coherence and objectivity). Only changes in the allocation of relative abundance between dominant and subordinate symbols seem to have true relevance for probability distributions to achieve the reference distribution or to deviate from it (as shown in Tables 1 and 2 of the submitted manuscript).

7 - Reviewer’s comment:

“I have reported plagiarism, which is self-plagiarism: Figure 1 is taken (copied as it is) from an earlier publication of the same author.”

Author’s response:

I must disagree with the reviewer’s comment. Actually to me is hard to understand why the reviewer has reported self-plagiarism regarding Figure 1 of the submitted manuscript.

In my previous publication about the Lorenz curve (Camargo, J.A. 2019. The Lorenz curve: a suitable framework to define satisfactory indices of landscape composition. Landscape Ecology 34: 2735-2742) I used the Lorenz curve as a graphical representation of patch (land-cover) dominance for thirteen hypothetical landscapes. The probability distributions of these thirteen hypothetical landscapes (I - XIII) are different from the probability distributions of the ten hypothetical messages (I - X) in Table 1 of the submitted manuscript that I have used to represent symbol dominance with the Lorenz curve in Figure 1 of the submitted manuscript. In fact, as we can see below, my published Figure 3 in Landscape Ecology is clearly different from my Figure 1 of the submitted manuscript. The only similarity between both figures is that the Lorenz curve is ideally depicted within a unit (1x1) square in which the 45-degree (diagonal) line represents perfect equality (either patch evenness or equiprobability).

8 - Reviewer’s comment:

 “I have assessed the topic as an extremely interesting topic, so the readers of Entropy would be interested in new results and new proposals for measuring dominance, diversity and information content of a sequence. But republishing an earlier result, without clear notice of this fact and just mentioning the original paper as a reference, is disappointing for an enthusiastic reader.”

Author’s response:

I must disagree with the reviewer’s comment. At the end of the ‘Introduction’ section of the submitted manuscript I clearly report that “I recently found that the corresponding versions of dC1, dC2, DC1 and DC2 exhibited better mathematical behavior than the corresponding versions of dS, dR, DS and DR when measuring land-cover dominance and diversity in hypothetical landscapes. This better mathematical behavior was inherent to the compatibility of dC1 and dC2 with the Lorenz curve-based graphical representation of land-cover dominance (see Camargo 2019).”

Again I have to insist that as far as I know no previously published paper has related the three concepts of symbol dominance, symbol diversity and information entropy, and much less within the framework of the Lorenz curve. Furthermore, as far as I know, no previously published paper has proposed Lorenz-consistent measures of information entropy such as HC1 = log2 DC1 = log2 (N – ∑(pd – ps)) and HC2 = log2 DC2 = log2 (N – ∑│pi – pj│). My overall conclusion is that the mathematical behavior of Camargo’s entropies (HC1 and HC2) is better than the mathematical behavior of Shannon’s entropy (HS = – ∑ pi log2 pi) and Rényi’s second order entropy (HR = log2 1/∑ pi2). The use of dC1, DC1 and HC1 is particularly recommended, as only changes in the allocation of relative abundance between dominant (pd > 1/N) and subordinate (ps < 1/N) symbols are of real relevance for probability distributions to achieve the reference distribution (pi = 1/N) or to deviate from it.

Reviewer 2 Report

The paper is well written and can be suitable for publication with a minor revision:

1-Can the author expresses the results in view of Bonferroni and Zenga curves?

2-Can we express in place of maximum vertical distance from Lorenz curve and motivate the assertions. 

Author Response

Responses to the comments of reviewer #2:

I appreciate the general comment of the reviewer #4. Before expressing my responses to the two specific comments, I want to point out that these responses are intended to satisfy the reviewer's apparent curiosity about what I can think of the potential use of the Bonferroni and Zenga curves to define satisfactory measures of symbol dominance, symbol diversity and information entropy. In fact, I assume the idea of the reviewer is not that I must use the Bonferroni and Zenga curves to develop measures of symbol dominance, symbol diversity and information entropy, since the core of my submitted manuscript is the Lorenz curve, comparing the mathematical behavior of novel Lorenz-consistent statistics with the mathematical behavior of statistics based on the classical Shannon and Rényi entropies.

1 - Reviewer’s comment:

“Can the author express the results in view of Bonferroni and Zenga curves?”

Author’s response:

The Lorenz curve (Lorenz 1905), the Bonferroni curve (Bonferroni 1930) and the Zenga curve (Zenga 2007) are graphical methods primarily used to analyze the inequality in the distribution of income among the individuals of a population. These three inequality curves are ideally depicted within a unit (1x1) square in which the cumulative proportion of income is related to the cumulative proportion of individuals. Although I realize that, owing to the increasing economic inequality during the present market globalization, some authors have supported the use of Bonferroni and Zenga curves and related indices to better assess poverty, as these inequality measures are oversensitive to lower levels of the income distribution, to me the Lorenz curve represents the best and most logical framework to define satisfactory indices of inequality (dominance) and associated measures of diversity/entropy. This personal feeling has been explicitly expressed at the end of the ‘Introduction’ section of the second (revised) version of the submitted manuscript (lines 77-79 of the revised manuscript with track changes). Indeed, I do not think that the results/conclusions of my submitted manuscript can be expressed in view of the Bonferroni and Zenga curves, due mainly to their significant differences with the Lorenz curve.

Since the Bonferroni and Zenga curves and related indices are oversensitive to lower levels of the income distribution, Bonferroni-based inequality estimates tend to be higher than Lorenz-based inequality estimates, and Zenga-based inequality estimates tend to be higher not only than Lorenz-based inequality estimates but also than Bonferroni-based inequality estimates (see, for example, Pasquazzi and Zenga. 2018. Components of Gini, Bonferroni, and Zenga inequality indexes for EU income data. Journal of Official Statistics, 34: 149-180). Other handicap is that the Bonferroni curve and related index do not obey the Dalton principle of replication invariance (see, for example, Giorgi. 2006. Concentration index, Bonferroni. Encyclopedia of Statistical Sciences). This replication principle is usually considered as fundamental of a satisfactory measure of inequality.

By contrast, the Lorenz curve and Camargo`s Lorenz-consistent statistics of symbol dominance (dC1 and dC2) obey the Dalton principle of replication invariance (as shown in Figure 1 and Table 3 regarding probability distributions I – V in Table 1 of the submitted manuscript). Also, the statistics of symbol dominance (dS and dR), that are based on the Shannon and Rényi entropies (HS and HR), obey the Dalton principle of replication invariance (as shown in Table 3 regarding probability distributions I – V in Table 1 of the submitted manuscript).

Therefore, it is likely that in the case of developing statistics of information entropy consistent with the Bonferroni and Zenga curves, those statistics may exhibit worse mathematical behavior than the classical Shannon and Rényi entropies (HS and HR) when measuring the amount of information or uncertainty in a probability distribution. However, in the submitted manuscript, I show that the novel Lorenz-consistent statistics of information entropy (HC1 and HC2) can quantify more efficiently than HS and HR the amount of information or uncertainty in a probability distribution, particularly when differences between higher and lower probabilities are maximized by increasing the number of small probabilities. After all, within the context of classical information theory, the information content of a symbol is an increasing function of the reciprocal of its probability.

2 - Reviewer’s comment:

“Can we express in place of maximum vertical distance from Lorenz curve and motivate the assertions.”

Author’s response:

Sorry, but I am not sure what your question means. I will however try to answer it as best as possible.

In the case of the  Lorenz curve-based graphical representation of symbol dominance (Figure 1 of the submitted manuscript), the 45-degree (diagonal) line represents equiprobability or perfect equality, symbol dominance (the extent of relative abundance inequality among different symbols, particularly between dominant and subordinate symbols) being then linked to the separation (convexity) of the Lorenz curve from the 45-degree line. This convex separation may be quantified as the maximum vertical distance from the Lorenz curve to the 45-degree line if only differences in relative abundance between dominant and subordinate symbols are of interest (a symbol is dominant because its probability (pd) > 1/N and is subordinate because its probability (ps) < 1/N, N being the number of different symbols in a message), or as twice the area between the Lorenz curve and the 45-degree line if differences in relative abundance among all symbols are of interest. The value of Camargo’s dominance statistic (dC1) is equivalent to the maximum vertical distance, and the value of Camargo’s dominance statistic (dC2) is equivalent to twice the area, both dominance statistics taking the same value whenever relative abundance inequality occurs only between dominant and subordinate symbols (as shown in Table 3 of the submitted manuscript). In this regard it is worth noting that Camargo’s dominance statistic (dC1) is comparable to the Schutz’s (1951) index of income inequality (also known as Pietra ratio or Robin Hood index) and Camargo’s dominance statistic (dC2) is comparable to the Gini’s (1914, 1921) index of income inequality (the Gini index and the Schultz index take the same value whenever income differences occur only between the rich and the poor).

In the case of the Bonferroni inequality curve the line of perfect equality is usually represented by the upper horizontal x-axis, the degree of inequality being linked to the area between the Bonferroni curve and the upper horizontal x-axis (i.e., the area above the Bonferroni curve). Since the Bonferroni inequality curve is not necessarily convex, and because the line of perfect equality is not represented by the 45-degree (diagonal) line, I do not think that an inequality measure similar to the maximum vertical distance in the Lorenz curve can be found for the Bonferroni inequality curve.

A similar assumption may be expressed regarding the Zenga inequality curve. In this case the line of perfect equality is usually represented by the lower horizontal x-axis, the degree of inequality being linked to the area between the Zenga curve and the lower horizontal x-axis (i.e., the area below the Zenga curve). Since the Zenga inequality curve is not necessarily convex, and because the line of perfect equality is not represented by the 45-degree (diagonal) line, I do not think that an inequality measure similar to the maximum vertical distance in the Lorenz curve can be found for the Zenga inequality curve.

Reviewer 3 Report

Please find the attached pdf. 

Author Response

Responses to the comments of reviewer #3:

I appreciate very much the general comments of the reviewer #3 that my submitted manuscript is “original, nicely written, and well structured. Existing literature is referred to at appropriate places throughout the manuscript. The details of the comparison carried out between different statistics are reported in tables. I appreciate the author very much for doing excellent work”.

I also appreciate the minor comments of the reviewer #3. My responses to those minor comments are:

1 - Reviewer’s comment:

“In the manuscript, the author has used a first-person singular pronoun, which is unusual in academic writing. For example, ‘…I derive novel measures….’, ‘…that I had previously proposed…’. In current days, even first-person plural pronoun is avoided for writing journal articles. The reason might be that when we use first-person pronouns, it shows somewhat self-serving, which is not the objective of the research. However, this comment has no relation to the quality of the manuscript, and the author/editor is to make the final decision.”

Author’s response:

Because it is a very personal paper regarding ideas, arguments and novel statistics, I have considered mandatory the use of the first-person singular pronoun. However, taking into account the reviewer’s comment, I have minimized this use in the second (revised) version of the manuscript. More specifically, the first-person singular pronoun has been removed from the lines 9, 62, 150, 391 of the revised manuscript with track changes).

2 - Reviewer’s comment:

“In ‘Abstract’, ‘Introduction’ and ‘Conclusions’, the author states that the comparison is made to show the proposed one is better. Are not the results a consequence of the method, not previously determined/predictable? Also, the ‘Introduction’ section should report only the objectives of research, not the result.”

Author’s response:

Probability distributions in Tables 1 and 2 of the submitted manuscript were selected to better assess differences in mathematical behavior between Lorenz-consistent statistics (Camargo’s indices) and Lorenz-inconsistent statistics (indices based on Shannon’s entropy and Rényi’s second order entropy). Otherwise, in the case of using probability distributions chosen at random, we can obtain results that do not allow us to appreciate significant differences between the respective mathematical behaviors. This explanation has been incorporated at the end of the subsection 2.3 (Comparing Lorenz-Consistent Statistics with HS-Based and HR-Based Statistics) of the second (revised) version of the submitted manuscript (lines 288-292 of the revised manuscript with track changes).

With the use of these specific probability distributions I have really demonstrated the existence of significant differences in mathematical behavior, Lorenz-consistent statistics exhibiting greater coherence and objectivity than Lorenz-inconsistent statistics. In fact, as shown in Tables 3 and 4 regarding probability distributions in Tables 1 and 2, the value of Camargo’s dominance statistic (dC1) always is equivalent to the whole relative abundance of dominant symbols that must be transferred to subordinate symbols to achieve equiprobability, that is, maximum coherence and objectivity because only changes in the allocation of relative abundance between dominant and subordinate symbols (but not among dominant symbols or among subordinate symbols) seem to have true relevance for probability distributions to achieve the reference distribution (involving equiprobability) or to deviate from it. Furthermore, as shown in Table 3 regarding data in Table 1, whereas the normalized entropies of HC1 and HC2 increase from distribution VII to X, the normalized entropies of HS and HR decrease markedly. And as commented in the section 3 (Results and Discussion), this remarkable finding would indicate that HC1 and HC2 can quantify more efficiently than HS and HR the amount of information or uncertainty in a probability distribution, particularly when differences between higher and lower probabilities are maximized by increasing the number of small probabilities. After all, within the context of classical information theory, the information content of a symbol is an increasing function of the reciprocal of its probability.

According to ‘Instructions for authors' of the scientific journal Entropy the author(s) should briefly mention the main aim of the work and highlight the main conclusions at the end of the ‘Introduction’ section. This is the reason because I have included the main result/conclusion of my work.

3 - Reviewer’s comment:

“Line 27: Should it not be ‘Shannon’s entropy’ instead of ‘Shannon’s information entropy’? Because Shannon’s entropy is based on information theory only”.

Author’s response:

I agree with the reviewer’s comment. The term ‘Shannon’s information entropy’ has been replaced with the simpler term ‘Shannon’s entropy’ in the second (revised) version of the manuscript (lines 28, 45 of the revised manuscript with track changes).

4 - Reviewer’s comment:

“The first sentence in the ‘Introduction’ section is too long that it may be difficult to read. Also, in some other places of the manuscript, the author has used long sentences. If possible, they may be shortened”.

Author’s response:

I agree with the reviewer’s comment. The first sentence in the ‘Introduction’ section has been rewritten and shortened in the second (revised) version of the manuscript (lines 28-34 of the revised manuscript with track changes).

5 - Reviewer’s comment:

“Line 143: ‘i’ and ‘j’ both are the indexes. Is it necessary to write the notation for pi or pj repeatedly?”

Author’s response:

The notations for pi and pj have been rewritten as “pi and pj are the relative abundances of two different symbols in the same message” in the second (revised) version of the manuscript (lines 166-167 of the revised manuscript with track changes).

6 - Reviewer’s comment:

“In some of the places in the manuscript, the author may use parentheses for writing the entropy function, e.g., log2 ( )”.

Author’s response:

The parentheses are used to express Camargo’s indices of information entropy HC1 = log2 DC1 = log2 (N – ∑(pd – ps)) and HC2 = log2 DC2 = log2 (N – ∑│pi – pj│). However, I do not consider it necessary to express Shannon’s entropy HS = – ∑ pi log2 pi and Rényi`s second order entropy HR = log2 1/∑ pi2.

7 - Reviewer’s comment:

“The captions of tables and figure may be shortened”.

Author’s response:

Sorry, but I consider that all the information contained in the captions is necessary to facilitate the understanding of tables and figure to potential readers.

Reviewer 4 Report

My personal assessment on the work is positive, owing to the relevant, timely and clarity of the study, as well as the focus of analysis. 

So, the study is interesting and results may have political implications.

But I believe it is necessary to undertake a set of modifications, in order to improve the current manuscript. The authors might wish to address these comments if they deem them to be important.

Below I explain them in detail.

The paper is divided into four parts: 1) Introduction; 2) Materials and Methods; 3) Results and Discussion; 4) Concluding Remarks.

But it is suggested that the addition of a subtitle in the introduction would draw attention to the paper outline for readers whose main interest is in the third and fourth sections of the paper. This suggestion is made to assist less technically interested readers.

The introduction and the conclusions is very short compared, for example, to the Research methods. I would write more background about the topic so who are not familiar with the topic and read the investigation can understand it better.

To expand citations from other current papers that reviews the literature.

Because of the lack of a proper current literature review the results are not discussed in the context of previous findings. In this sense, the theoretical and empirical underpinnings of the model specification are at best can be improved .

It is necessary to expand the working hypotheses put forward and the specific research problems. This suggestion is made to assist less technically interested readers.

More information is required on the definition of the indicators selected and need further explanation in relation to the period covered by these indicators.

To correct this, the authors need to include more theoretical underpinnings to their work and draw out the policy implications of the results much more clearly.

Because of the lack of a proper current literature review the results are not discussed in the context of previous findings.

Some practitioner readers of Entropy wish to concentrate on the findings of the paper and the recommendations for policy. The policy implications of the analysis need to be made much more clear. 

Author Response

Responses to the comments of reviewer #4:

I appreciate very much the general comments of the reviewer #4 that “the work is positive, owing to the relevant, timely and clarity of the study, as well as the focus of analysis”, but I am not sure what the reviewer means by the sentence “results may have political implications”.

I also appreciate the reviewer’s suggestions to improve the submitted manuscript. My responses to those suggestions are:

1 - Reviewer’s comment:

“The paper is divided into four parts: 1) Introduction; 2) Materials and Methods; 3) Results and Discussion; 4) Concluding Remarks. But it is suggested that the addition of a subtitle in the introduction would draw attention to the paper outline for readers whose main interest is in the third and fourth sections of the paper. This suggestion is made to assist less technically interested readers.”

Author’s response:

Sorry, but I consider that a subtitle in the ‘Introduction’ section is not necessary, as the submitted manuscript is a specific theoretical study and relatively short. In the case of a review paper could however be necessary.

2 - Reviewer’s comment:

“The introduction and the conclusions is very short compared, for example, to the Research methods. I would write more background about the topic so who are not familiar with the topic and read the investigation can understand it better.”

Author’s response:

The ‘Introduction’ section has been extended by incorporating background information about the traditional use of the Lorenz curve in the field of econometrics to graphically analyze the inequality in the distribution of income among the individuals of a population, the popular use of Lorenz-consistent statistics of income inequality (such as the Gini and Schutz indices) in econometrics, and the more recent use of other graphical methods (such as the Bonferroni and Zenga curves) in econometrics to analyze income inequality (lines 70-79 of the revised manuscript with track changes). Also, additional explanations have been incorporated to the ‘Concluding Remarks’ section to support the conclusions (lines 393-403 of the revised manuscript with track changes).

3 - Reviewer’s comment:

“To expand citations from other current papers that reviews the literature.”

Author’s response:

The following ten additional references have been incorporated into the second (revised) version of the submitted manuscript, as the ‘Introduction’ section has been extended:

  1. Gini, C. Sulla misura della concentrazione e della variabilità dei caratteri. Atti del Reale Istituto Veneto di Scienze, Lettere ed Arti 1914, 73, 1203-1248.
  2. Gini, C. Measurement of inequality of incomes. Econ. J. 1921, 31, 124-126.
  3. Schutz, R.R. On the measurement of income inequality. Am. Econ. Rev. 1951, 41, 107-122.
  4. Idrees, M.; Ahmad, E. Measurement of income inequality: a survey. FJES 2017, 13, 1-32.
  5.  
  6. Milanovic, B. Global Inequality: A New Approach for the Age of Globalization; The Belknap Press of Harvard University Press: Cambridge (MA), USA, 2016.
  7. Bonferroni, C. Elementi di Statistica Generale; Libreria Seeber: Firenze, Italy, 1930.
  8. Zenga, M. Inequality curve and inequality index based on the ratios between lower and upper arithmetic means. Stat. Appl. 2007, 5, 3-27.
  9. Giordani, P.; Giorgi, G.M. A fuzzy logic approach to poverty analysis based on the Gini and Bonferroni inequality indices. Stat. Method. Appl. 2010, 19, 587-607.
  10. Greselin, F.; Pasquazzi, L.; Zitikis, R. Contrasting the Gini and Zenga indices of economic inequality. J. Appl. Stat. 2013, 40, 282-297.
  11. Pasquazzi, L; Zenga, M. Components of Gini, Bonferroni, and Zenga inequality indexes for EU income data. J. Off. Stat. 2018, 34, 149-180.

4 - Reviewer’s comment:

“Because of the lack of a proper current literature review the results are not discussed in the context of previous findings. In this sense, the theoretical and empirical underpinnings of the model specification are at best can be improved.”

Author’s response:

As far as I know no previously published paper has related the three concepts of symbol dominance, symbol diversity and information entropy, and much less within the framework of the Lorenz curve. Moreover, as indicated in the ‘Introduction’ section of the submitted manuscript, novel measures of symbol dominance (dC1 and dC2), symbol diversity (DC1 = N (1 – dC1) and DC2 = N (1 – dC2)) and information entropy (HC1 = log2 DC1 = log2 (N – ∑(pd – ps)) and HC2 = log2 DC2 = log2 (N – ∑│pi – pj│)) are derived from Lorenz-consistent statistics that I had previously proposed to quantify dominance and diversity in community ecology (Camargo 1992a, 1992b, 1993, 1995, 2008) and landscape ecology (Camargo 2019). This is the reason because the obtained results are not discussed in the context of previous findings of other authors. However, the obtained results are discussed regarding the traditional use of Shannon’s (1948) entropy (HS = – ∑ pi log2 pi) and Rényi`s (1961) second order entropy (HR = log2 1/∑ pi2) in classical information theory, also comparing the mathematical behavior of Lorenz-inconsistent measures (i.e., HS-based and HR-based statistics of dominance, diversity and entropy = dS, dR, DS, DR, HS and HR) with the mathematical behavior of Lorenz-consistent measures (i.e., Camargo’s statistics of dominance, diversity and entropy = dC1, dC2, DC1, DC2, HC1 and HC2). All this has been more extensively explained in the second (revised) version of the submitted manuscript.

5 - Reviewer’s comment:

“It is necessary to expand the working hypotheses put forward and the specific research problems. This suggestion is made to assist less technically interested readers.”

Author’s response:

The three working assumptions/hypotheses have been clearly explained in the ‘Introduction’ section of the second (revised) version of the submitted manuscript (lines 51-55). In addition, because the ‘Introduction’ section has been extended by incorporating background information about the use of the Lorenz curve and Lorenz-consistent inequality indices in econometrics, the aims of the work result more clear and evident to less technically interested readers.

6 - Reviewer’s comment:

“More information is required on the definition of the indicators selected and need further explanation in relation to the period covered by these indicators.”

“To correct this, the authors need to include more theoretical underpinnings to their work and draw out the policy implications of the results much more clearly.”

Author’s response:

Sorry, but I cannot understand the meaning of these reviewer’s comments.

In the submitted manuscript I do not select any indicator that covers any period. I first use the Lorenz curve as the key framework to assess symbol dominance, symbol diversity and information entropy. Subsequently, I derive novel measures of symbol dominance (dC1 and dC2), symbol diversity (DC1 and DC2) and information entropy (HC1 and HC2) from Lorenz-consistent statistics that I had previously proposed to quantify dominance and diversity in community ecology (Camargo 1992a, 1992b, 1993, 1995, 2008) and landscape ecology (Camargo 2019). The contrast between symbol dominance and symbol redundancy is also highlighted.

On the other hand, I do not draw out any policy implication from the obtained results because I do not realize whether the results may have policy implications. In my opinion the answer to this question would involve great subjectivity and speculation, which would detract from the scientific relevance of the submitted manuscript.

7 - Reviewer’s comment:

“Because of the lack of a proper current literature review the results are not discussed in the context of previous findings.”

Author’s response:

Since this comment is the same as the previous comment 4, my response is also the same.

8 - Reviewer’s comment:

“Some practitioner readers of Entropy wish to concentrate on the findings of the paper and the recommendations for policy. The policy implications of the analysis need to be made much more clear.”

Author’s response:

Sorry, but again I must insist that I do not realize whether the obtained results may have policy implications, and therefore cannot make any recommendation for policy.

Round 2

Reviewer 1 Report

The author has considered all my questions and comments. The literature is well extended with more relevant references. The issue of self plagiarism is cleared as well. I had access only to not so well quality copy of the figure, and at that quality, the two figures were not distinguishable. Now, I am convinced, that the results, presented here, are original and not published earlier.

The manuscript has been improved in sense of discussing the role of the numerical examples. Now it is correctly discussed in the response, that the advantages of the presented measures are evident only in the special cases analysed here. 

If the author would include the following statement from the response in the manuscript:

"in the case of using probability distributions chosen at random, we can obtain results that do not allow us to appreciate significant differences between the respective mathematical behaviors"

the paper would be acceptable.

Author Response

I appreciate very much comments and clarifications of the reviewer #1 about my responses to the reviewer’s previous comments (round 1).

I appreciate very much comments and clarifications of the reviewer #1 about my responses to the reviewer’s previous comments (round 1).

Regarding the reviewer`s new comment (round 2) that “if the author would include the following statement from the response in the manuscript: “in the case of using probability distributions chosen at random, we can obtain results that do not allow us to appreciate significant differences between the respective mathematical behaviors” the paper would be acceptable”, I must point out that this statement was already included in the revised manuscript: lines 290-292 in Entropy-751286 with track changes; lines 272-274 in Entropy-751286 without track changes.